# The impact of preovulatory versus midluteal serum progesterone level on live birth rates during fresh embryo transfer

**Abdelhamid Benmachiche**[1]*, **Sebti Benbouhedja**[1], **Abdelali Zoghmar**[1], **Peter Samir Hesjaer Al Humaidan**[2,3]

**1** Center for Reproductive Medicine, Ibn-rochd, Constantine, Algeria, **2** The Fertility Clinic, Skive Region Hospital, Skive, Denmark, **3** Department of Clinical Medicine, Aarhus University, Aarhus C, Denmark

* benmachiche@gmail.com, benmachiche@gmail.com

**Data Availability Statement:** Dataset supporting the findings of the current study is available in the Supporting Information files.

## Abstract

### Background

Conflicting evidence still prevails concerning the effect of preovulatory elevated progesterone ($EP_4$) on reproductive outcomes in fresh embryo transfer (ET). However, few studies have analyzed the effect of $EP_4$ on the likelihood of pregnancy using multivariate regression approach. The potential confounding factors tested in these studies were limited to either patient's characteristics or to stimulation related parameters. Yet, several studies have shown that postovulatory parameters such as midluteal progesterone ($P_4$) level may be considered as a proxy variable of endometrial receptivity as well.

### Objective

The aim of the present study was to estimate the independent effect of preovulatory $P_4$ effect, if any, on the probability of live birth (LB) by considering the midluteal endocrine profile when controlling for the potential confounding factors.

### Methods

This is a secondary data analysis of a cohort of fresh IVF/ICSI cycles triggered with GnRH agonist (n = 328) performed in a single IVF center during the period 2014–2016. Patients contributed only one cycle and were stratified into four groups according to preovulatory $P_4$ quartiles. We assessed the association between preovulatory $P_4$ and the odds of LB calculated by logistic regression analysis after controlling for the most clinically relevant confounders. The primary outcome measure: Live birth rates (LBR).

### Results

Both preovulatory and midluteal $P_4$ were significantly correlated with the ovarian response. Logistic regression analysis showed that preovulatory serum $P_4$ did not have a significant impact on LBR. In contrast, midluteal serum $P_4$ level was an important independent factor

**Funding:** The author(s) received no specific funding for this work.

**Competing interests:** The authors have declared that no competing interests exist.

associated with LBR. The optimal chance of LBR was achieved with midluteal serum $P_4$ levels of 41–60 ng/ml, [OR: 2.73 (1.29–5.78); p< 0.008].

## Conclusion

The multivariate analysis suggests that the midluteal $P_4$ level seems to impact LBR more than the preovulatory $P_4$ level in women undergoing IVF treatment followed by fresh ET.

## Introduction

During the natural cycle, the progesterone ($P_4$) rise precedes the luteinizing hormone (LH) peak on the day of ovulation [1], and thus, may play a physiological role in the ovulation process [2]. After ovulation, $P_4$ is essential for the secretory transformation of the endometrium and, thus, for the implantation [3,4]. However, during controlled ovarian hyperstimulation, 6–30% of cycles may display a rise in serum $P_4$ levels particularly at the end of stimulation mostly higher than those of natural cycle [5,6]. The preovulatory $P_4$ rise might be attributed to an increased number of follicles, each one produces a physiological amount of $P_4$, rather than $P_4$ being produced by granulosa cells as a consequence of premature luteinization [7]. Over the years, different thresholds of serum $P_4$ have been proposed ranging from 0.8 to 2 ng/mL above which deleterious in vitro fertilization (IVF) outcome may occur during fresh embryo transfer (ET) [5,6]. Although the impact—or not of elevated progesterone ($EP_4$) on reproductive outcomes has been debated for almost three decades [7–9], this controversy remains unsolved, presumably due to the lack of well-designed studies. In this respect, strong reservations have recently been expressed regarding methodological approaches applied to address this question, indicating that live birth (LB) is a multifactorial process that cannot be defined by a single threshold value, in most instances suggested arbitrarily after dichotomizing continuous data [10–12]. Further, the weakness of bivariate analysis in this context has also been well documented through receiver operating characteristic (ROC) curves as reported in several previous studies demonstrating that the predictive performance of the preovulatory $P_4$ level to discriminate between conception and non-conception cycles is very limited [13–15]. Accordingly, the implementation of prognostic prediction models instead of simple bivariate analysis is mandatory to address the issue and thus, might serve as a useful tool to estimate the relative contribution of different factors to a single outcome [11,16]. Indeed, the multivariable analysis provides the ability to remove the effect of confounders or other forms of biases and, thus, get a more realistic picture compared to looking at only one variable [17–19]. So far, only a few studies performed multivariate regression analyses to explore the possible effect of $EP_4$ on the likelihood of pregnancy [11,20–23]. Interestingly, the covariates tested in the regression models of these studies were mostly related either to the patient's characteristics or to the ovarian stimulation parameters.

To our knowledge, none of the above studies incorporated parameters related to the luteal phase particularly the endocrine profile, presumably owing to a lack of the availability of serum hormones measurements, as the hormonal assessment is not routinely performed during the luteal phase following fresh embryo transfer. Yet prior investigations have shown that the midluteal $P_4$ level may also be considered as a promising predictor of IVF outcome, not only in fresh but also in frozen-thaw transfer cycles [24,25]. Based on this evidence, we hypothesized that by controlling for the differences in midluteal $P_4$ levels, the effect of the preovulatory serum $P_4$ on the probability of LB might be more accurately estimated. Thus, the primary

objective of the present study was to estimate the independent effect of preovulatory $P_4$, if any, on the probability of LB after 26 completed gestational weeks by considering the midluteal $P_4$ level when controlling for the most common confounding factors. Moreover, the relationship between preovulatory and midluteal $P_4$ was investigated.

## Materials and methods

### Study design

This is a secondary analysis of data from a cohort of 328 patients undergoing a fresh IVF/intra-cytoplasmic sperm injection (ICSI) cycles in which GnRH agonist (GnRH-a) was used for triggering final oocyte maturation. The details of the flow chart of participants in the study have been described previously [26]. Each patient was only included once. All cycles were performed at the IVF unit- Ibn Rochd, Constantine, Algeria, between 2014 and 2016. All patients gave written informed consent to participate in the study, which was conducted in accordance with the Declaration of Helsinki and Good Clinical Practice. The research project was approved by the Ethics Committee of the University hospital Centre Ibn Badis, Constantine, 20 October 2013. The original study was registered in ClinTrial.gov, Number: 02053779.

### Ovarian stimulation

In brief, ovarian stimulation was performed with GnRH antagonist co-treatment, using exclusively recombinant follicle stimulating hormone (r-FSH), (Gonal F., Merck Serono; Puregon., MSD) The initial dose of r- FSH was individualized for each patient according to the female's age and ovarian reserve markers (basal FSH level antral follicular count). Stimulation was started from the day 3 of menstruation and dose adjustments were performed on day 5 or 6 of stimulation based on ovarian response. GnRH antagonist 0.25 mg was daily administered from cycle day 5 or 6 when follicles reached a mean diameter of 13 mm and continued until to the day of final oocyte maturation. The ovulation was triggered with GnRH-a when minimum of two follicles reached 17 mm or more in size and followed by oocyte pick up (OPU) 36–38 hours later. Fertilization of mature oocytes was carried out using either conventional IVF or ICSI technique based on the sperm quality. Blood collection was done for FSH, LH and Estradiol ($E_2$) on day 1 of stimulation, for FSH, LH, $E_2$ and $P_4$ on the day of triggering and on day OPU+7 and for Beta-human chorionic gonadotropin (ß-hCG) on day OPU + 14. Serum was analyzed for endocrine parameters by a central laboratory.

Blood samples have been processed according to the manufacturer's instructions. Serum was analyzed immediately using a Vidas kit (BioMerieux, France). The calibration range of the VIDAS Progesterone kit is 0.25–80 ng/mL.

### Embryo grading and embryo transfer

One to three embryos per patient were replaced on day OPU+2 or 3, depending upon the age and the ovarian response. Embryo quality was assessed at the cleavage stage based on the embryo morphology. A good quality embryo (Grade 1 and 2) was defined as follows: the 2–4 cells on day 2 and 6–8 cells by day 3, <20% of fragmentation, and regular shaped cells [27]. No embryo transfer was cancelled due to $EP_4$ on the day of GnRH-a trigger.

### Luteal phase support

Luteal phase support (LPS) was previously described with more details [26]. Briefly, all patients received a bolus of HCG 1500 IU 1 hour after OPU, micronized progesterone vaginally (600 mg/day) and estradiol orally (4 mg/day) starting from the night of OPU and continuing until

12 weeks of gestation or a negative pregnancy test. Besides, patients have been randomized on the day of embryo transfer into two groups; the study group received an additional single dose of GnRH-a (Triptorlin 0.1 mg) on day OPU+6 while the control group did not.

The primary outcome of the current study was live birth rate (LBR) defined as a live neonate beyond 26 weeks of gestation. Further, the correlation between preovulatory serum $P_4$ levels, midluteal serum $P_4$ levels and ovarian response was investigated.

### Statistics

All statistical analyses were performed using IBM SPSS Statistics 26.0 (IBM Inc., New York, USA).

The distributions of continuous parameters were evaluated using the Shapiro–Wilk test to determine whether each variable followed a normal distribution. Serum $P_4$ levels on the day of trigger as well as midluteal $P_4$ (OPU+7) levels were converted from continuous variables into categorical variables by apportioning them into four groups (quartiles) based on $25^{th}$, $50^{th}$ and $75^{th}$ percentiles. Q1 included 0–25%, Q2 included 25–50%, Q3 included 50–75% and Q4 included 75–100%. Data are presented as means and standard deviations for continuous data with normal distribution, as medians and ranges for continuous data with skewed distribution and as percentages for categorical variables.

Differences in skewed continuous data between the four preovulatory $P_4$ groups were assessed using Kruskal–Wallis test followed by a post hoc pairwise comparison in case of a statistical difference between groups. One-way analysis of variance analysis (Anova) was used in case of normal continuous data. Difference in categorical variables between $P_4$ groups was assessed using Pearson's chi-square test or Fishers exact test where appropriate. Spearman's correlation coefficient was used to assess the association between preovulatory and midluteal $P_4$ levels as well as ovarian response elements in terms of $E_2$ on the trigger day, number of follicles $> 11$ mm and number of oocytes retrieved. Patients contributed only one cycle in the dataset analysed. Logistic regression analysis was used to assess 13 parameters possibly related to LBR, including covariates demonstrating a $P \leq .25$ for the association with outcome in the univariable models as well as clinically relevant predictive variables which were selected based on previous studies [11,21,28]. The factors tested in the model were: (i) *Patient's characteristics*: female age, female BMI and number of previous failed IVF (ii) *The intensity of ovarian response*: total dose of FSH consumption, duration of stimulation, number of follicles $> 11$ mm on the day of trigger which, optimally, equals to the number of oocytes. (iii) *LH on the day of GnRH-a trigger* (iv) *Embryo's characteristics*: number of embryos obtained, whether at least one good embryo transferred and number of transferred embryos. Additionally, an adjustment for midluteal $P_4$ along with LPS imbalances by including the extra dose of GnRH-a (yes versus no) as a covariate was performed. A standard, i.e., direct logistic regression was used as an analysis method to develop the final model [29]. Box and Tidwell, 1962 procedure was assessed to test the linearity of the continuous variables with respect to the logit of the dependent variable, i.e., LB [30]. The multicollinearity among all the factors was examined using the variance inflation factor (VIF). The model fit was evaluated by the Hosmer and Lemeshow test [18]. Odds ratios (ORs) and 95% confidence intervals (CIs) were assessed distinctly for each factor. All statistical analyses were two-tailed, and results were considered significant when p-values $< 0.05$ were obtained.

## Results

### Baseline patient and cycle characteristics

The present study evaluated a total of 328 IVF/ICSI cycles followed by fresh ET. The spectrum of patients was considered as large including both normal responders (NR): 80% (260/328)

**Table 1. Baseline patient and cycle characteristics according to serum $P_4$ quartiles on the day of GnRH agonist trigger.**

| Characteristic [a] | $P_4$ Q1 (<0.74) | $P_4$ Q2 (0.75–0.94) | $P_4$ Q3 (0.95–1.30) | $P_4$ Q4 (>1.30) | P-value [b] | Total |
|---|---|---|---|---|---|---|
| Number | 88 | 80 | 79 | 81 | NA | 328 |
| Age (years) | 32.50 ±3.66 | 32.25 ± 3.84 | 31.68 ±3.77 | 31.17±4.02 | .12 | 31.91±3.84 |
| BMI (kg/m₂) | 28.69 ±5.17 | 28.45 ± 4.31 | 27.52 ±4.72 | 27.23±4.19 | .12 | 27.99±4.65 |
| Previous IVF, (n) | 1 (1–3) | 1 (1–3) | 1 (1–3) | 1 (1–4) | .44 | 1.3 ± 0.57 |
| Stimulation, (days) | 9 (7–15) | 9 (6–12) | 9 (6–13) | 9 (6–15) | .79 | 9(6–15) |
| r-FSH (IU) | 1871.30±333.63 | 1825.93±294.50 | 1809.17±288.54 | 1840.43±303.25 | .66 | 1837.65±305.71 |
| Follicles >11mm trigger, (n) | 9 (4–28) | 10 (5–26) | 15 (4–30) | 16 (4–26) | .0001 | 13 (4–30) |
| E2 trigger, (pg/mL) | 1277.50(37–6298) | 1646.45(304–3000) | 1880(426–3000) | 2600(433–4300) | .0001 | 1943.21 (304–6298) |
| LH trigger, IU/L | 0.92 (0.10–3.30) | 0.98 (0.10–4.65) | 0.93 (0.10–4.57) | 1.25 (0.10–6) | .02 | 1.28 (0.10–6) |
| $P_4$ (OPU+7), ng/mL | 38.71 (10–127) | 36.20 (7–182) | 40 (14–322) | 49 (11.78–192.80) | .02 | 45.53 (7–322) |
| $E_2$ (OPU+7), pg/mL | 739(89–4747) | 903(110–4300) | 948(144–6071) | 1033(182–4315) | .001 | 867.47 (89–6071) |
| LH (OPU+7), IU/L | 2.5(0.10–13.43) | 1.93(0.10–9) | 1.60(0.10–8.32) | 2.47(0.10–9.34) | .003 | 2.77 (0.10–13.43) |
| FSH (OPU+7), IU/L | 0.74 (0.14–2) | 0.67 (0.10–2.46) | 0.63 (0.15–1.81) | 0.68 (0.19–2.50) | .41 | 0.77 (0.10–2.50) |

[a] Descriptive data are presented as mean ± SD for continuous normal data and as median (range) for continuous skewed data. Groups are compared using Anova or Kruskal-Wallis tests as appropriate.

[b] Two-side P < .05 were considered significant.

BMI, body mass index; E2, estradiol; IVF, in vitro-fertilization; IU, international units; LH, luteinizing hormone; NA, not applicable; OPU, oocyte pick-up; $P_4$, progesterone(ng/ml); Q, quartile; r-FSH, recombinant follicle-stimulating hormone; SD, standard deviation.

and high responders (HR) >18 follicles: 20% (68/328). Baseline and cycle characteristics according to quartiles of serum $P_4$ levels on the day of trigger are provided in Table 1. The overall mean female age and female body mass index (BMI) were 31.17 ± 4.02 years and 27.23 ± 4.19 kg/m², respectively. Patients were divided into four distinct groups according to their quartile serum $P_4$ levels on the day of GnRH-a trigger: [Q1: <0.74, Q2: 0.75–0.98, Q3: 0.99–1.30, and Q4: > 1.30 ng/mL]. Conversion factor to SI unit, 3.180. The four groups (Q1, Q2, Q3 and Q4) were comparable as regards age, BMI, duration of ovarian stimulation, total dose of FSH and serum FSH on day OPU+7. However, there were significant differences between the groups regarding the ovarian response parameters (number of follicles >11mm and $E_2$) as well as hormones on day OPU+7 (LH, $E_2$ and $P_4$).

## Relationship between preovulatory, midluteal $P_4$ levels and ovarian response

Spearman's correlation revealed that the preovulatory $P_4$ level was significantly correlated with ovarian response elements in terms of $E_2$ levels, number of follicles >11 mm and number of oocytes retrieved (All $P<0.0001$) as well as with midluteal $P_4$ level ($P<0.007$) (S1 Table).

## Reproductive outcomes

Reproductive outcomes are provided in Table 2. The overall positive hCG rate per transfer, ongoing pregnancy rate and LBR in the study was 44.2% (145/328), 34.5% (113/328) and 33.5% (110/328), respectively. Although the number of oocytes retrieved as well as the number of embryos obtained were significantly different between the different $P_4$ groups on the day of ovulation trigger, however, the pregnancy outcomes were comparable (Table 2).

Table 3. summarizes the results of a multivariate regression analysis related to the LBR. The logistic regression model was statistically significant, X2 = 60.02, p < .0005. The model explained 23.6% (Nagelkerke R2) of the variance in LB and correctly classified 73% of cases.

**Table 2. Outcome of ovarian stimulation, fertilization and embryo transfer according to serum P4 quartiles on the day of GnRH agonist trigger.**

| Characteristic[a] | P4 Q 1 (<0.74) | P4 Q2 (0.75–0.94) | P4 Q 3 (0.95–1.30) | P4 Q 4 (>1.30) | P value[b] | Total |
|---|---|---|---|---|---|---|
| Number | 88 | 80 | 79 | 81 | NA | 328 |
| Oocytes retrieved | 8.47 ± 4.88 | 7.78 ± 3.74 | 10.33 ± 5.84 | 11.33 ±5.84 | .0001 | 9.41 ± 4.97 |
| 2PN oocytes | 5.43 ± 3.42 | 5.08 ± 2.60 | 5.94 ± 3.43 | 6.46 ± 3.43 | .04 | 5.72 ± 3.27 |
| Embryos | 5.09 ± 3.24 | 4.79 ± 2.62 | 5.59 ± 3.14 | 6.17 ± 3.33 | .02 | 5.41 ± 3.13 |
| Embryo transfer | 2.31 ± 0.63 | 2.43 ± 0.61 | 2.30 ± 0.58 | 2.32 ± 0.49 | 0.50 | 2.34 ± 0.58 |
| Positive hCG, n (%) | 41 (46.60%) | 34 (42.5%) | 38 (48.10%) | 32 (39.50%) | 0.68 | 145 (44.2%) |
| Ongoing pregnancy, n (%) | 31 (35.22%) | 30 (37.5%) | 28 (35.44%) | 24 (29.63%) | 0.74 | 113 (34.5%) |
| Live birth, n (%) | 31 (35.22%) | 29 (36.25%) | 26 (32.91%) | 24 (29.62%) | .41 | 110 (33.5%) |

aAll values are presented as mean ± (SD) or count n (%)

bKruskal-Wallis test or Chi-squared test for differences between preovulatory serum P4 groups.

E2, estradiol; IU, international units; LH, luteinizing hormone; NA, not applicable; OPU, oocyte pick-up; P4, progesterone (ng/mL); PN, pronuclei; Q, quartile; r-FSH, recombinant follicle-stimulating hormone; SD, standard deviation.

**Table 3. Multivariate regression analysis of independent factors related to the live birth.**

| Variable | Regression coefficient | Standard error | OR | 95% CI | P-value |
|---|---|---|---|---|---|
| P4 day of trigger, ng/mL | | | | | |
| Q1 < 0.74 [reference category] | | | 1 | | |
| Q2 (0.75–0.94) | .004 | .366 | 1.00 | 0.49–2.06 | 0.99 |
| Q3 0.95–1.30) | -.038 | .372 | 0.96 | 0.46–2.00 | 0.92 |
| Q4 >1.30 | -.582 | .392 | 0.56 | 0.26–1.20 | 0.14 |
| LH day of trigger, IU/L | | | | | |
| Q1 < 0.68 [reference category] | | | 1 | | |
| Q2 (069–0.98) | .178 | .379 | 1.19 | 0.57–2.51 | 0.64 |
| Q3 (0.99–1.60) | .206 | .375 | 1.23 | 0.59–2.56 | 0.58 |
| Q4 >1.60 | .856 | .379 | 2.35 | 1.12–4.94 | 0.02[a] |
| Follicles> 11 mm day of trigger, (n) | | | | | |
| (1–6) [reference category] | | | 1 | | |
| (7–18) | -.434 | .391 | 0.65 | 0.30–1.40 | 0.29 |
| (>18) | -2.263 | .586 | 0.10 | 0.03–0.33 | 0.000[a] |
| Embryos, (n) | .142 | .054 | 1.15 | 1.04–1.28 | 0.008[a] |
| Embryos transferred, (n) | .621 | .294 | 1.86 | 1.04–3.31 | 0.03[a] |
| P4 (OPU+7), ng/mL | | | | | |
| Q1 <28 [reference category] | | | 1 | | |
| Q2 (29–40) | -.397 | .379 | 0.67 | 0.32–1.41 | 0.29 |
| Q3 (41–60) | 1.007 | .382 | 2.73 | 1.29–5.78 | 0.008[a] |
| Q4 >60 | .406 | .453 | 1.50 | 0.62–3.64 | 0.37 |

For estimates, adjustment was made for female age; female BMI; number of previous failed IVF; duration of stimulation; total dose of FSH during stimulation; number of follicle > 11mm on the day of trigger; hormones on the day of trigger (P4 and LH); number of embryos obtained; number of embryos transferred; whether at least one good embryo transferred; additional dose of GnRH agonist on day OPU+6 (yes vs. no) and P4 on day OPU+7.

Note

Preovulatory serum P4 was compared between the first quartile (<0.74 ng/mL; reference category). and the rest of quartiles (2–4). Follicles> 11 mm were compared between low ovarian response (< 6 follicles; reference category), and intermediate response (6–18) and high ovarian response (>18). Preovulatory serum LH was compared between the first quartile (<0.68 IU/L; reference category), and the rest of quartiles (2–4). Midluteal serum P4 (OPU+7) was compared between the first quartile (<28 ng/mL; reference category) and the rest of quartiles (2–4).

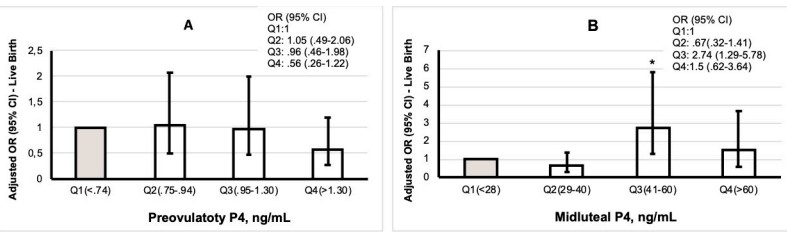

* P < 0.05; CI: confidence interval; OR: odds ratio; P4: progesterone; Q (Q1-Q4) quartiles; Q1: reference category

**Fig 1.** Forest plot of adjusted live birth rates according to preovulatory (A) versus midluteal serum $P_4$ levels (B) after IVF/ICSI with fresh ET. Data are presented in OR (95% CI) of the comparison of the odds between each $P_4$ quartile with the lowest $P_4$ quartile (Q1: reference category). For estimates of live birth, adjustment was made for female age; female BMI; number of previous failed IVF; duration of stimulation; total dose of FSH during stimulation; number of follicle > 11mm on the day of trigger; hormones on the day of trigger ($P_4$ and LH); number of embryos obtained; number of transferred embryos; whether at least one good embryo transferred; additional dose of GnRH agonist on day OPU+6 (yes vs. no) and $P_4$ on day OPU+7.

Sensitivity was 43.1%, specificity was 88.3%, positive predictive value was 65.3% and negative predictive value was 75.2%. The independent factors found to be significantly associated with LB were: midluteal serum $P_4$ level, i.e., on day OPU+7, serum LH levels and final follicles >11 mm on the day of trigger, number of embryos obtained, number of transferred embryos and whether at least one good embryo transferred. However, adding the extra dose of GnRH agonist on day OPU+6 (yes versus no) to the regression model did not change estimates significantly.

Fig 1 depicts the OR, 95% CI for LBR according to serum $P_4$ quartiles on the day of ovulation trigger versus serum $P_4$ quartiles on day OPU+7. Each group of $P_4$ was compared with the lowest quartile (Q1) [Fig 1A and 1B], respectively. After adjustment for relevant confounders, serum $P_4$ on the day of trigger did not have a significant impact on LBR [Fig 1A]. In contrast, the LBR increased significantly in patients with a midluteal serum $P_4$ level of 41–60 ng/mL compared to the lowest quartile Q1 ($P_4 < 28$ ng/mL = a reference category); [OR: 2.73 (1.29–5.78); p< 0.008] [Fig 1B].

## Discussion

Using a multivariate regression analysis, the present study suggests that the preovulatory P4 level did not show a significant effect on reproductive outcomes. In contrast, the midluteal $P_4$ level significantly impacted the LBR in a nonlinear pattern suggesting that both low and high $P_4$ levels during this period, may reduce the chance of LB in women undergoing IVF treatment followed by fresh ET. Indeed, an optimal midluteal $P_4$ range (41–60) ng/mL was identified. The same pattern was seen for both crude and adjusted OR of preovulatory and midluteal $P_4$ levels. Furthermore, a positive correlation was found between the magnitude of the ovarian response and $P_4$ levels on the day of ovulation induction as well as on the day OPU +7. Regarding the preovulatory $P_4$, our data failed to reproduce the findings of previous studies, particularly those which implemented a multivariate regression analysis and showing negative effect of $EP_4$ on reproductive outcomes [11,20–26]. The discrepancy between these studies and our findings may be attributed to the following reasons; (i) Aside from the hypothesis of being a true null result, an underpowered genuine effect of preovulatory $P_4$ on LBR may be considered in the framework of the present study design limitations since the sample size is relatively modest compared to the previous reports which may constitute a type II error [31]. (ii) Another possibility that deserves to be examined in the interpretation of the present results is that we included the midluteal $P_4$ as a novel covariate to adjust for along with the patient

characteristics and stimulation parameters; this may generate different estimates in the regression model. (iii) The current published data on $EP_4$ and IVF outcomes predominantly derive from hCG triggered cycles [8] whereas in our study, all patients were triggered with GnRH-a. In this respect, the $P_4$ concentration during the early luteal phase was found to be significantly higher in hCG trigger compared to GnRH-a trigger [32,33] which may yield substantial discrepancy in the endocrine profile [34].

In contrast with earlier studies suggesting that the impact of $EP_4$ on pregnancy outcome does not seem to be modulated by the ovarian response [20,21], it has been recently shown that the number of oocytes may be an important confounder associated with both the exposure ($EP_4$) and the outcome (LB) [11]. In this respect, accumulating evidence revealed that $EP_4$ does not seem to significantly affect reproductive outcomes in the high response (HR) category in which $EP_4$ is being more common than poor and intermediate response categories. [11,14,23]. In the current study, the final follicle count >11mm was found in the regression analysis to significantly reduce LBR in HR (> 18 follicles) when compared with poor and intermediate responders [OR: 0.10, 95% CI, 0.03–0.33] (Table 3), which is consistent with the findings reported by a recent study showing a steady reduction in LBR beyond twenty oocytes retrieved, presumably due to supraphysiological circulating steroids levels [35]. Besides, the present data show a significant correlation between the ovarian response and both preovulatory and midluteal $P_4$ levels (S1 Table) suggesting that each follicle contributes to the pool of serum $P_4$ before ovulation triggering [20,36] as well as after oocyte retrieval [24,37]. Contrasting the preovulatory serum $P_4$ level, the current results suggest that the midluteal $P_4$ level is an independent factor associated with LB potential, and, interestingly, in a nonlinear pattern, [OR: 2.73 (1.29–5.78); $p< 0.008$] (Table 3). While sufficient evidence has accrued, demonstrating that a low luteal $P_4$ level is associated with low pregnancy rates in fresh ET despite transfer of morphologically good embryos [38], the impact of $EP_4$ during the midluteal period on the cycle outcome has not been fully elucidated. A recent study analyzing a dataset of 602 patients reported that the optimal chance of pregnancy was achieved with midluteal serum $P_4$ of 150–250 nmol/l, i.e., 47–78 ng/mL which is close to the optimal range found in our study, 41–60 ng/mL [24].

Currently, it is not yet clarified whether the late follicular $P_4$ rise is a cause or a confounder of lower reproductive outcome in fresh ET since the conclusions drawn from bivariate analysis may be prone to bias [11]. Further, in the majority of previous studies where the midluteal $P_4$ was not considered in the multivariate regression analysis, it is believed that the preovulatory $EP_4$ per se reduces pregnancy outcome by altering the endometrial receptivity [39,40] rather than oocyte/embryos quality [41].

Conversely, the current study argues against the above-mentioned findings, showing that the detrimental effect of $EP_4$ seems to be attributed to the $EP_4$ during midluteal time rather than to $EP_4$ at the day of ovulation induction. As depicted in Fig 1A, preovulatory $P_4$ levels did not affect LBR. Importantly, the nonlinear model of the correlation between midluteal $P_4$ levels and LBR suggests that there is an optimal range of midluteal $P_4$, a window at which the most optimal implantation rates can be expected since pregnancy rates are lower with both low and high $P_4$ levels Fig 1B, which is in agreement with a previous report [42]. Indeed, the present study suggests that a suboptimal midluteal $P_4$ level seems to decrease the chance of LB following fresh ET from 45.10% in patients with optimal midluteal $P_4$ levels (41–60) ng/mL to 30.12%, 25.27% and 30.77% in patients with $P_4 < 28$ ng/mL, $P_4$ (29–40) ng/mL and $P_4 >60$ ng/mL respectively (S2 Table). Accordingly, we speculate that the likely reason for the negligible effect of preovulatory $EP_4$ on the cycle outcome of HR may be explained by the availability of adequate $P_4$ around the time of implantation rather than the availability of high-quality embryos including blastocysts for transfer, as previously

suggested [43,44]. Further, one might wonder whether reports failing to show any relation between late follicular $P_4$ levels and the reproductive outcome may have analyzed more HR who had an appropriate midluteal endocrine profile. Providing further support for our observations, Wang et al. retrospectively compared the effect of preovulatory versus early luteal $P_4$ levels on reproductive outcomes in 384 patients [34]. The study showed that low responders undergoing intensive ovarian stimulation are more likely to exhibit low reproductive outcomes in fresh ET compared to normal and HR which was attributed to an elevated $P_4$ ratio (the rise in $P_4$ between trigger day and oocyte retrieval) rather than to preovulatory $P_4$ level per se.The strength of the present findings derives from the fact that they are provided by multivariable regression analysis instead of bivariate analysis which may remove the effect of confounders and, thus, more reliably may estimate the true effect of preovulatory $P_4$ on the LBR. Nonetheless, the study also has some limitations, including the fact that the observed data derive from a post hoc analysis which may prevent statistical detection of further significant differences particularly when the study is underpowered and not truly negative; moreover, data collection ended in 2016; at that time, the use of blastocyst transfer was not systematically implemented in our center. Although new confounders such as those related to the luteal phase have been added to the regression model, there could still be residual variables which are not considered. Furthermore, the findings of the current study may not be applicable to a population triggered with hCG. Lastly, the validity of a single measurement instead of a median value [45], and the performance and precision of immunoassay systems, particularly in the lower range of detectable $P_4$levels (<2.5 ng/mL) [46]. From a clinical perspective, this study highlights that monitoring $P_4$ levels during the midluteal phase would be an important and innovative practice which might increase the reproductive outcomes of fresh ET; thus, in patients with $P_4$ levels below 40 ng/mL, additional progesterone could still be provided to rescue the cycle [47]; alternatively in cycles with $P_4$ levels above 60 ng/mL, additional progesterone support is redundant and likely to be harmful for the endometrial receptivity. To conclude, the findings of the present study suggest that the midluteal $P_4$ level seems to impact LBR more than the preovulatory $P_4$ level. The most optimal midluteal $P_4$ level identified was 41–60 ng/mL, and both preovulatory and midluteal phase serum $P_4$ are positively correlated with the ovarian response. More research into the luteal phase steroids profile of the IVF cycle is needed before final conclusions can be drawn.

## Supporting information

**S1 Table. Spearman's correlation between preovulatory, midluteal serum $P_4$ concentration and ovarian response.** *N, 328 patients undergoing IVF/ICSI treatment. Rho, Spearman's correlation coefficient. P < .05: Statistically significant.*
(DOC)

**S2 Table. Relationship between Live birth rates and midluteal (OPU+7) serum $P_4$ quartiles.** *Chi-squared test for differences between mid-luteal serum $P_4$ groups. P < .05: Statistically significant. NA, not applicable; OPU, Ovum pick-up; $P_4$, progesterone (ng/mL).*
(DOC)

**S1 Dataset.**
(DOC)

**S2 Dataset.**
(XLSX)

## Acknowledgments

The authors thank Abdelmadjid Barkat and Abdelhamid Aberkane as a member and a president of a Medical Research Ethics Committee (University Hospital Benbadis, Constantine) respectively, for their valuable collaboration.

## Author Contributions

**Conceptualization:** Abdelhamid Benmachiche, Peter Samir Hesjaer Al Humaidan.

**Data curation:** Abdelhamid Benmachiche, Peter Samir Hesjaer Al Humaidan.

**Formal analysis:** Abdelhamid Benmachiche, Peter Samir Hesjaer Al Humaidan.

**Methodology:** Abdelhamid Benmachiche.

**Supervision:** Sebti Benbouhedja, Abdelali Zoghmar, Peter Samir Hesjaer Al Humaidan.

**Validation:** Sebti Benbouhedja, Abdelali Zoghmar, Peter Samir Hesjaer Al Humaidan.

**Writing – original draft:** Abdelhamid Benmachiche, Sebti Benbouhedja, Abdelali Zoghmar, Peter Samir Hesjaer Al Humaidan.

**Writing – review & editing:** Abdelhamid Benmachiche, Peter Samir Hesjaer Al Humaidan.

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
