## [Decision Letter · Decision Letter 0]

16 Nov 2020

PONE-D-20-32335

The preovulatory versus the midluteal serum progesterone level: Which is better for live birth prediction during fresh embryo transfer?

PLOS ONE

Dear Dr. Benmachiche,

Thank you for submitting your manuscript to PLOS ONE. After careful consideration, we feel that it has merit but does not fully meet PLOS ONE’s publication criteria as it currently stands. Therefore, we invite you to submit a revised version of the manuscript that addresses the points raised during the review process.

The reviewers have raised several methodological and statistical issues that require addressing

We look forward to receiving your revised manuscript.

Kind regards,

Stephen L Atkin, MD

Academic Editor

PLOS ONE

Journal Requirements:

3. Thank you for submitting the above manuscript to PLOS ONE. During our internal evaluation of the manuscript, we found significant text overlap between your submission and the following previously published works, some of which you are an author.

https://academic.oup.com/humrep/article/17/4/933/644620

https://fjfsdata01prod.blob.core.windows.net/articles/files/474821/pubmed-zip/.versions/1/.package-entries/fendo-10-00639/fendo-10-00639.pdf?rscd=attachment%3B+filename%2A%3DUTF-8%27%27fendo-10-00639.pdf&se=2019-10-29T16%3A29%3A43Z&sig=5dOv7luvnmpO%2BCpvLrsI%2FQ6r02D5F9cQPZzP8gX5A1s%3D&sp=r&sr=b&sv=2015-12-11

Please revise the manuscript to rephrase the duplicated text, cite your sources, and provide details as to how the current manuscript advances on previous work. Please note that further consideration is dependent on the submission of a manuscript that addresses these concerns about the overlap in text with published work.

Reviewers' comments:

Reviewer's Responses to Questions

**Comments to the Author**

1. Is the manuscript technically sound, and do the data support the conclusions?

Reviewer #1: Yes

Reviewer #2: No

2. Has the statistical analysis been performed appropriately and rigorously? 

Reviewer #1: Yes

Reviewer #2: No

3. Have the authors made all data underlying the findings in their manuscript fully available?

Reviewer #1: Yes

Reviewer #2: No

4. Is the manuscript presented in an intelligible fashion and written in standard English?

Reviewer #1: Yes

Reviewer #2: Yes

5. Review Comments to the Author

Reviewer #1: The present study aims to evaluate the impact of preovulatory and midluteal progesterone levels on live birth rates in the course of IVF. In this attempt 328 IVF cycles were evaluated. The authors concluded that midluteal progesterone levels were associated with LBR. Indeed, progesterone ranges between 41 and 60 ng/mL were found to be related with higher pregnancy potential and live birth chances.

The research question is of interest and falls within the scope of the journal. The study is well-designed and the manuscript is written concisely and in standard English.

However, there are some issues that deserve consideration:

In contrast to previous publications preovulatory progesterone was not associated with pregnancy potential. Please provide a power analysis to ensure that the sample size was sufficient to exclude a type II error.

The number of embryos transferred was shown to influence results. Please report whether midluteal progesterone levels were associated with implantation rates (i.e. singletons versus multiples) and may, thus, have contributed to progesterone levels.

Though the reviewer highly cherishes the scientific level of the discussion, the number of references could be tightened.

Reviewer #2: 1) The term “nonparametric variables” is awkward. The term “nonparametric” can be used to describe modeling methods, but not variables.

2) It is unclear what “A sequential logistic regression was used as an analysis method to develop the final model” means. Please elaborate. Does “sequential” means “stepwise” https://en.wikipedia.org/wiki/Stepwise_regression ?

3) Please use “multiple logistic regression model” to correctly describe the regression models used.

4) From Table 3, it is still unclear to me what models were fitted and what is the final model. Please explicitly clarify what variables / models were considered, how the variable selection was performed, what is the final model.

5) All the sentences on prediction performance, e.g, ROC, should be removed, because there is only one dataset involved in the current study. One cannot evaluate the prediction performance of a model on the same dataset based on which the model is derived.

6. PLOS authors have the option to publish the peer review history of their article (what does this mean?). If published, this will include your full peer review and any attached files.

Reviewer #1: **Yes: **Andrea Weghofer, MD

Reviewer #2: No

---

## [Author Response · Author response to Decision Letter 0]

22 Dec 2020

Dear Editor,

We appreciate the opportunity to revise and resubmit our paper titled “The preovulatory versus the midluteal serum progesterone level: Which is better for live birth prediction during fresh embryo transfer? “ 

We want to thank the editor and reviewers for their constructive comments and criticism helping us improving the article. We provided a point-by-point response to each of the editor’s and reviewers’ comments. We have included the page and line numbers in the revised manuscript to help the reviewers identify our changes. 

Reviewer’s comments are written in italic; authors’ responses are shown in upright font. 

We believe that we had addressed all the questions and concerns raised by the editor as well as reviewers. 

Should you have any further requests or questions, please do not hesitate to contact me.

Abdelhamid Benmachiche

Corresponding Author 

RESPONSE TO COMMENTS FROM THE EDITOR AND REVIEWERS

I. Editor

According to your comments and suggestions, we have provided changes as follows:

Manuscript format 

We have made the requested formatting adjustments to comply with the PLOS ONE manuscript checklist.

Regarding the figure extension, we have uploaded our figure file (Fig1) to the Preflight Analysis and Conversion Engine (PACE) digital diagnostic tool, https://pacev2.apexcovantage.com and get the label with the extension TIF (Fig1.tif)

Text similarity

We went over the parts of our manuscript that show similarity with previous literature, and rephrased the content to the extent possible accompanied by an appropriate citation. 

Seen from the same perspective, that is, to avoid redundancy and copyright issues, we have also removed the supplementary figure 1 (S1_Fig) which describes the flow chart of patients taken from our own original study (1) and we subsequently referred the reader to the source.

Please see page 4 line (70, 71) in the revised paper

 (1): Reference number [26] in the revised manuscript

Title: The dos and don’ts about how to write a great title from PLOS ONE 

suggest that the title – in the most cases- shouldn’t need to be framed as a question 

 (https://plos.org/resource/how-to-write-a-great-title/)

 Accordingly, the title of our manuscript has been modified 

 Original: 

 “The preovulatory versus the midluteal serum progesterone level: Which is better for 

 live birth prediction during fresh embryo transfer? ” 

 Revised:

 “The Impact Of Preovulatory Versus Midluteal Serum Progesterone Level On Live 

 Birth Rates During Fresh Embryo Transfer” 

 � Please see page 1 in the revised paper

Ethics statement 

 Written informed consent with more details has been added in the Methods and online 

 submission information

 � Please see page 4, lines (72, 74) in the revised paper

Data Availability Statement:

 The authors confirm that the data supporting the findings of this study are available as 

 a part of manuscript supporting information without restriction.

Reviewer 1

Question #1

In contrast to previous publications preovulatory progesterone was not associated with pregnancy potential. Please provide a power analysis to ensure that the sample size was sufficient to exclude a type II error.

Response #1

In the present study, we found that preovulatory P4 does not impact the LBR which is in line with many previous publications [5, 6, 7, 13]. However, in the discussion section, we thought necessary to focus rather on the discrepancy between our findings and those studies reporting a negative effect of elevated preovulatory P4 on pregnancy outcome [20-22].

Regarding the post-hoc power, we would say that:

We appreciate the reviewer’s insightful suggestion and agree that it would be useful to provide the study’s power in order to discriminate between a study which is truly negative or simply underpowered, however, we were not able to compute a post-hoc power for explaining the observed data, but we could only estimate it because its retrospective nature. 

While the utility of prospective power analysis in experimental design is universally accepted, the usefulness of retrospective techniques is controversial (1). 

This was also nicely explained by Hoenig and Heisey ( 2001) indicating that there is a large current literature that advocates the inappropriate use of post-experiment power calculations as a guide to interpreting tests with statistically nonsignificant results. The authors show that, power is mathematically directly related to the p-value; hence, calculating power once you know the p-value associated with a statistic is of little help in interpreting results (2). So, we will always have low observed power when we report non-significant effects.

In fact, the data of the current study were handled using a sample size which has been already calculated in the original study with a power set before starting the enrollment of patients (3). However, in this post-hoc data analysis, i.e., using the same sample size, we found that there was no significant result regarding preovulatory P4 levels on the live birth rate, then - by definition - its power to detect the effect actually observed is low for this parameter (Preovulatory P4).

So, without a priori power and sample size calculations, one can never be sure if the results obtained were due to a ‘‘true’’ effect (2)

For this reason, we did not claim that our results are truly negative as that was clearly acknowledged in the discussion section (type II error), because it is possible that the true effect size, if any, is even smaller.

Please see page 13, lines (263-266) in the revised paper 

On the other hand, the impact of differences in the midluteal P4 levels with the same sample size show a significant effect on LBR, thus, one may conclude that the study was appropriately powered for this variable, i.e., midluteal P4. 

For all these reasons and for the complex calculation of several sample sizes related to every variable examined, we considered unwise to perform a post hoc analysis.

Furthermore, we are confident that a statistically significant p value still can offer a reliable and useful information even if the power analysis has not been performed.

Nevertheless, we recognize that your comment should be mentioned in the paper as a limitation, so we added the following sentence “Nonetheless, the study also has some limitations, including the fact that the observed data derive from a post hoc analysis which may prevent statistical detection of further significant differences particularly when the study is underpowered and not truly negative”

Please see page 15, lines (322-324) in the revised paper.

References

(1) Thomas, L. (1997) Retrospective power analysis. Conservation Biology 11(1):276–280

(2) Hoenig and Heisey (2001). The Abuse of Power. The American Statistician 55(1):19-24

(3) Benmachiche A, Benbouhedja S, Zoghmar A, Boularak A, Humaidan P. Impact of mid-luteal phase GnRH agonist administration on reproductive outcomes in GnRH agonist-trigger: a randomized controlled trial. Front Endocrinol. (2017) 8:12 doi: 10.3389/fendo.2017.00124.

Question #2

The number of embryos transferred was shown to influence results. Please report whether midluteal progesterone levels were associated with implantation rates (i.e. singletons versus multiples) and may, thus, have contributed to progesterone levels.

Response #2

Thanks for pointing out this comment.

The ‘implantation rate’ (IR) is calculated as [IR = ngestational sacs/ntransferred embryos ]. This calculation can be performed per patient or aggregated per group of patients.

In the current study the IR according to midluteal P4 was calculated per quartiles groups and the results are provided below:

Implantation Rates According To Midluteal P4 Groups

 Q1 P4 (OPU+7) Q2 P4

 (OPU+7) Q3 P4 

(OPU+7) Q4 P4 (OPU+7) P-value

Nb Gestational Sacs 44 48 72 28 

Nb Embryo transferred 199 213 234 121 

Implantation rate n, (%) 44/199 (22.11%) 48/213

(22.53%) 72/234

 (30.77%) 28/121

(23.14) NS

Besides, the Pearson´ correlation showed a significant positive correlation between the number of embryos transferred (ET) and the number of gestational sacs (GS) (P<.01).

Although the mean number of ET (mean +/- SD: 2.3 +/- .58, P= .60) was similar between the four groups, however, a bivariate analysis found that the IR was higher in the third quartile (Q3: 72/234 (30.77%)) compared with the rest of quartiles, Q1: 44/199 (22.11%), Q2: 48/213 (22.53%) and Q4: 28/121 (23.14%), but the difference did not reach the significant level. Consequently, the midluteal P4 (41-60 ng/ml) threshold appears to enhance the implantation rate. 

Thereafter, in terms of LB, we found the same trend of results importantly with significant level always in favor of the third quartile of midluteal P4 in both bivariate and multivariate regression analysis (S1_Table), (Table 3) respectively, presumably due to the quality of implanted embryos.

Question #3

Though the reviewer highly cherishes the scientific level of the discussion, the number of references could be tightened.

Response #3

We agree with your suggestion that the number of references in the discussion section could be tightened.

Accordingly, we have removed 12 references

Original:

Number of references: 59

Revised:

Number of references: 47

Please see page 16, line 362

The list of the removed references includes:

References numbers: 8, 23, 26, 28, 29, 31, 37, 44 ,46, 48, 55 and 58 from the original version.

We deeply value the input of reviewer 1 

Reviewer 2

Question #1

The term “nonparametric variables” is awkward. The term “nonparametric” can be used to describe modeling methods, but not variables.

Response #1

We apologise for our lack of clarity. Accounting for the given suggestion, the term “nonparametric variables” has been changed to ” skewed continuous data “

In the revised manuscript.

Revisions in text are made for more clarity including specific information:

“Data are presented as means and standard deviations for continuous data with normal distribution, as medians and ranges for continuous data with skewed distribution and as percentages for categorical variables. Differences in skewed continuous data between the four preovulatory P4 groups were assessed using Kruskal–Wallis test followed by a post hoc pairwise comparison in case of a statistical difference between groups. One-way analysis of variance analysis was used in case of normal continuous data”.

Please see page 6, lines (117-122) in the revised manuscript

We hope that it is now clearer.

Question #2

It is unclear what “A sequential logistic regression was used as an analysis method to develop the final model” means. Please elaborate. Does “sequential” means “stepwise” https://en.wikipedia.org/wiki/Stepwise_regression ?

Response #2

We would like to thank the reviewer 2 who helped find the error and for giving us

this opportunity to clarify this important point.

Actually, the sentence “A sequential method” is wrong and was inadvertently typed by the corresponding author while the statistician team did not detect the error when checking the final draft, therefore it should be changed to “A standard/ direct method”. 

We have now made the following revisions:

Original: 

“A sequential logistic regression was used as an analysis method to develop the final model.

Revised: 

 “A standard, i.e., a direct method was used in logistic regression to develop the final model”.

The changes can be now seen in page 6, lines (138,139) of the revised paper.

Definitely we agree with the reviewer 2 and now we understand why the reviewer 2 was concerned by a discordance between the sentence “A sequential method” erroneously typed and the whole description of the model output as it was clearly stated in the questions #2 and #3.

Accordingly, the results provided in (Table 3) remain unchanged in the revised manuscript.

We apologize for our error and for this inconvenience.

Questions #3

Please use “multiple logistic regression model” to correctly describe the regression models used.

Response #3

You are absolutely right, the question #3 follows the question #2;

for the reason we mentioned above, when logistic regression uses “sequential method” in SPSS software, the output should logically exhibiting “multiple logistic regression models”, however, when the method used is a “standard/direct” which is labeled “Enter” in SPSS as we did to analyze our data we get in the final analysis only one model from which we can extract and display the factors showing a significant effect on the dependent variable.

Questions #4

From Table 3, it is still unclear to me what models were fitted and what is the final model. Please explicitly clarify what variables / models were considered, how the variable selection was performed, what is the final model.

Response #4

Similarly, the question #4 also follows question #2; since we performed a direct (i.e., full, standard, or simultaneous) logistic regression which is a default of sorts, since it enters all independent variables into the model at the same time and makes no assumptions about the order or relative worth of those variables, we will have only one final model.

Please see page (10,11), lines (208-229) of the revised paper

Questions #5

All the sentences on prediction performance, e.g, ROC, should be removed, because there is only one dataset involved in the current study. One cannot evaluate the prediction performance of a model on the same dataset based on which the model is derived.

Response #5

As per the request from the reviewer 2, all sentences on prediction performance have been removed from the revised manuscript.

We are grateful to the reviewer 2 again for the help and the constructive criticism.

---

## [Decision Letter · Decision Letter 1]

8 Jan 2021

PONE-D-20-32335R1

The Impact Of Preovulatory Versus Midluteal Serum Progesterone Level On Live Birth Rates During Fresh Embryo Transfer

PLOS ONE

Dear Dr. Benmachiche,

Thank you for submitting your manuscript to PLOS ONE. After careful consideration, we feel that it has merit but does not fully meet PLOS ONE’s publication criteria as it currently stands. Therefore, we invite you to submit a revised version of the manuscript that addresses the points raised during the review process.

please address the reviewers comment and resubmit

We look forward to receiving your revised manuscript.

Kind regards,

Stephen L Atkin, MD

Academic Editor

PLOS ONE

Reviewers' comments:

Reviewer's Responses to Questions

**Comments to the Author**

1. If the authors have adequately addressed your comments raised in a previous round of review and you feel that this manuscript is now acceptable for publication, you may indicate that here to bypass the “Comments to the Author” section, enter your conflict of interest statement in the “Confidential to Editor” section, and submit your "Accept" recommendation.

Reviewer #2: (No Response)

2. Is the manuscript technically sound, and do the data support the conclusions?

Reviewer #2: Yes

3. Has the statistical analysis been performed appropriately and rigorously? 

Reviewer #2: Yes

4. Have the authors made all data underlying the findings in their manuscript fully available?

Reviewer #2: No

5. Is the manuscript presented in an intelligible fashion and written in standard English?

Reviewer #2: Yes

6. Review Comments to the Author

Reviewer #2: 1) Please remove some remaining sentences on prediction analysis—lines 124-125 in Statisticis section; the second sentence in the Discussion section.

7. PLOS authors have the option to publish the peer review history of their article (what does this mean?). If published, this will include your full peer review and any attached files.

Reviewer #2: No

---

## [Author Response · Author response to Decision Letter 1]

15 Jan 2021

Dear Editor

We appreciate the effort and the time that you and the reviewers have dedicated to providing your valuable feedback on our manuscript. 

We believe that we had addressed all the questions and concerns raised by the editor as well as the reviewer #2. 

We have included the page and line numbers in the revised manuscript to help the reviewer identify our changes. 

We look forward to hearing from you regarding our revision submission and to respond to any further questions and comments you may have.

Abdelhamid Benmachiche (AB)

Corresponding Author 

RESPONSE TO COMMENTS FROM THE EDITOR AND REVIEWER #2

I. Editor

Data availability

According to The PLOS Data Policy, the authors are required to make all data underlying the findings described in their manuscript fully available without restriction. Three main options are being recommended; the data should be provided as part of the manuscript or its supporting information, or deposited to a public repository.

With your permission and for clarification, we take this opportunity to emphasize that the authors confirm that the relevant data supporting the findings of this manuscript are fully available in its supporting information files without any restriction;

S3 Dataset (XLSX)_File 1 provides data within a spreadsheet (Excel format).

S4 Dataset (DOCX)_File 2 (Word format) gives additional details such as the variables names, the variables meanings, the measurement units and the missing data which might be helpful for the analysis of data included in S3 Dataset (XLSX)_File 1.

Please see Supporting information files, page 16 lines 349 and 350 of the revised manuscript.

Thus, we believe that we have totally complied with the PLOS Data Policy

II. Reviewer #2

Question #1

Please remove some remaining sentences on prediction analysis—lines 124-125 in Statisticis section; the second sentence in the Discussion section.

Response #1

We completely agree with this and have, accordingly, incorporated the requested changes, that is, all sentences on prediction analysis have been removed from the revised manuscript.

.

Please see Statistic section, page 6 line 126 and Discussion section, page 12 lines 249-251 of the revised manuscript.

We deeply appreciate the reviewer’s input and insightful comments

---

## [Editor Report · Decision Letter 2]

20 Jan 2021

The Impact Of Preovulatory Versus Midluteal Serum Progesterone Level On Live Birth Rates During Fresh Embryo Transfer

PONE-D-20-32335R2

Dear Dr. Benmachiche,

We’re pleased to inform you that your manuscript has been judged scientifically suitable for publication and will be formally accepted for publication once it meets all outstanding technical requirements.

Kind regards,

Stephen L Atkin, MD

Academic Editor

PLOS ONE
---

## [Editor Report · Acceptance letter]

25 Jan 2021

PONE-D-20-32335R2 

The Impact Of Preovulatory Versus Midluteal Serum Progesterone Level On Live Birth Rates During Fresh Embryo Transfer 

Dear Dr. Benmachiche:

I'm pleased to inform you that your manuscript has been deemed suitable for publication in PLOS ONE. Congratulations! Your manuscript is now with our production department. 

Kind regards, 

on behalf of

Dr. Stephen L Atkin 

Academic Editor

PLOS ONE